# Chitosan-ZnO Nanocomposites Assessed by Dielectric, Mechanical, and Piezoelectric Properties

**DOI:** 10.3390/polym12091991

**Published:** 2020-09-01

**Authors:** Evgen Prokhorov, Gabriel Luna-Bárcenas, José Martín Yáñez Limón, Alejandro Gómez Sánchez, Yuriy Kovalenko

**Affiliations:** 1Cinvestav, Unidad Querétaro, Querétaro 76230, QRO, Mexico; gabriel.luna@cinvestav.mx (G.L.-B.); jmyanez@cinvestav.mx (J.M.Y.L.); alejandrogomez@cinvestav.mx (A.G.S.); 2Postgraduate Department, University of Aeronautics of Querétaro, Querétaro 76278, QRO, Mexico; kovalenko.yuriy@gmail.com

**Keywords:** chitosan, zinc oxide nanoparticles, interfacial layer, dielectric spectroscopy

## Abstract

The aim of this work is to structurally characterize chitosan-zinc oxide nanoparticles (CS-ZnO NPs) films in a wide range of NPs concentration (0–20 wt.%). Dielectric, conductivity, mechanical, and piezoelectric properties are assessed by using thermogravimetry, FTIR, XRD, mechanical, and dielectric spectroscopy measurements. These analyses reveal that the dielectric constant, Young’s modulus, and piezoelectric constant (d_33_) exhibit a strong dependence on nanoparticle concentration such that maximum values of referred properties are obtained at 15 wt.% of ZnO NPs. The piezoelectric coefficient d_33_ in CS-ZnO nanocomposite films with 15 wt.% of NPs (d_33_ = 65.9 *pC/N*) is higher than most of polymer-ZnO nanocomposites because of the synergistic effect of piezoelectricity of NPs, elastic properties of CS, and optimum NPs concentration. A three-phase model is used to include the chitosan matrix, ZnO NPs, and interfacial layer with dielectric constant higher than that of neat chitosan and ZnO. This layer between nanoparticles and matrix is due to strong interactions between chitosan’s side groups with ZnO NPs. The understanding of nanoscale properties of CS-ZnO nanocomposites is important in the development of biocompatible sensors, actuators, nanogenerators for flexible electronics and biomedical applications.

## 1. Introduction

Zinc oxide nanoparticles (ZnO-NPs) are one of the most attractive materials due to their unique optical, piezoelectric, mechanical, and antibacterial properties. Nanocomposites based upon ZnO-NPs are widely used for the development of different optoelectronic, electronic, sensors, collar cells, etc., devices (see, for example [1,2,3]). Recently, there were published significant publications about the potential use of ZnO-NPs that include flexible devices such as supercapacitance [4], flexible piezoelectric nanogenerators with ZnO-polyvinylidene fluoride (PVDF) [5,6], piezoelectric vibration sensors based on polydimethylsiloxane (PDMS) and ZnO nanoparticle [7], soft thermoplastic material with polyurethane matrix [8], poly (ethylene oxide) and poly (vinyl pyrrolidone) blend matrix incorporated with zinc oxide (ZnO) nanoparticles for optoelectronic and microelectronic devices [9], gate transistors with ZnO and ethyl cellulose [10], chitosan-ZnO (CS-ZnO) nanocomposite for packing applications [11,12,13], CS-ZnO as antibacterial agent [13,14,15], and CS-ZnO nanocomposite for supercapacitor [16].

Based upon the above information, chitosan-based nanocomposites offer significant scientific and technological potential. In this regard, chitosan (CS), a polysaccharide obtained from the deacetylation of chitin, is a natural polymer with high absorption capacity, biodegradability, biocompatibility with antibacterial features. Additionally, chitosan is a hydrophilic polymer with NH_2_ and OH side groups which can interact with ZnO nanoparticles via hydrogen bonding and form nanocomposites with new properties [17,18,19]. It is noteworthy that the literature reports publications that deal with different methods of CS-ZnO nanocomposite preparation and their antibacterial, optical, photocatalytic activity (see, for example [11,12,13,14,15,16,17,18,19,20]), and mechanical [15,21,22,23] properties. In the case of mechanical properties, it has been reported that the ZnO content improves mechanical properties not only in CS-ZnO composite [15,22,23] but also in CS-cellulose-ZnO [24] and in CS-PVA-ZnO [25] materials.

However, to the best our knowledge, the literature does not properly address the influence of ZnO content on the conductivity of CS-ZnO nanocomposite; in this regard there are two controversial articles related to the effect of ZnO additional on the dielectric constant of CS-ZnO membranes (in [23] with additional of ZnO NPs dielectric constant increase and in [26] decrease). The conductivity and dielectric properties play important role in applications of CS-ZnO nanocomposites in flexible organic electronics in a wide range of devices like transistors, sensors, flexible piezoelectric nanogenerators, ultraviolet photodetectors, photodiodes, etc., [4,23].

One of the most important questions related not only for application of CS-ZnO nanocomposites but also for all nanotechnology is how to find the best/optimum concentration of NPs with the best performance for different applications. CS-ZnO membrane composites consist of dielectric CS matrix and ZnO semiconductor fillers with wide bandgap with static dielectric constant ca. 8.5 [27] and low conductivity (ca. 10^−4^–10^−5^ S/cm, which depends upon intrinsic defects created by oxygen vacancies [28,29,30]). Therefore, this material can be considered as a dielectric matrix with dielectric inclusions. It is noteworthy that for different polymer-ZnO NPs composites, the dielectric constant depends upon ZnO content where a maximum is observed; for instance, in PVDF-ZnO at 0.06 vol.% of ZnO [31]; in PVDF-ZnO 5.5 vol.% of ZnO [32]; in PVDF-ZnO at 15 wt.% [33]: in PVA/PVP-ZnO at 8 wt.% [34]; in PVA-ZnO at 10 mol% [35]. The explanation of the maximum in the dielectric constant proposed in these articles is based upon classical percolation theory; by increasing conductivity inclusions in dielectric matrix the conductivity of composites increases at the percolation threshold and upon higher concentration of fillers there appears a saturation. The dielectric constant also shows a maximum near the percolation threshold [36,37]; however, refs. [31,32,33,34,35] do not report a conductivity percolation effect. It is noteworthy that ZnO NPs exhibit low conductivity such that the PVA-ZnO composite’s conductivity is ca. 10^−7^–10^−9^ S/cm [35] and PVA/PVP-ZnO is less than 10^−7^ S/cm [34]. Therefore, such material cannot exhibit conductivity percolation phenomena and this model cannot be used to explain the maximum in dielectric constant.

Similarly, to the dielectric behavior, there is a maximum on the Young’s modulus as a function of ZnO concentration. Ref. [38] reports a maximum on the Young’s modulus in PHBV-ZnO (Poly(3-hydroxybutyrate-co-3-hydroxyvalerate-ZnO) at a composition of 4 wt.% of NPs; Ref. [39] reports that system PEEK-ZnO (poly(ether ether ketone)-ZnO) shows a maximum at 5 wt.% of ZnO; PMMA-ZnO at 1 wt.% of NPs [40]; PLA-ZnO at 2 wt.% [41]. In summary, such a maximum in Young’s modulus has been related to the distribution of nanoparticles within the polymer matrix and strong interfacial adhesion that can enhance the mechanical properties of nanocomposites [42].

In general, in polymer-ZnO NPs composites both dielectric constant and Young’s modulus share a common feature: by increasing NPs concentration both properties increase [43,44]. Consequently, it is conceivable that dielectric, mechanical, and piezoelectric properties can be optimized by varying the concentration of ZnO NPs in chitosan nanocomposites for different applications.

Based upon the above discussion, this work aims to investigate the structural properties of CS-ZnO films including their dielectric, conductivity, mechanical and piezoelectric properties by varying the concentration of ZnO nanoparticles. To assess this study, we take advantage of impedance spectroscopy, FTIR, XRD, thermogravimetry, and piezoelectric measurements.

## 2. Materials and Methods

Chitosan (CS, medium molecular weight, deacetylation ca. 72%), acetic acid (99.7%), and ZnO NPs dispersed in water (20 wt.% in water) with dimension ca. 40 nm were purchased from Sigma Aldrich (St. Louis, MO, USA) and used as received.

CS solution (1 wt.%) was prepared in acetic acid solution (1 vol.%) and stirred for 24 h. Different amounts of ZnO sonicate colloidal solutions with various weight percent of ZnO (5, 10, 15, and 20 wt.% with respect to CS dry-base) were dispersed in the CS solution by ultrasound for 30 min at 60 Hz. Finally, 18 mL of each nanocomposite solutions were placed in Petri dishes and dried during 20 h at 60 °C to obtain films with thickness ca. 40 µm. For impedance measurements, CS-ZnO films were gold-sputtered on both sides to serve as contacts.

The amount of free water was determined by thermogravimetric analysis (TGA) (TGA 4000—PerkinElmer, Walham, MA, USA). Measurements were made in the dry air with a heating rate of 10 °C/min. The interaction between CS functional groups with ZnO was analyzed by FTIR measurements on a Perkin Elmer Spectrum GX spectrophotometer using ATR (MIRacle™) sampling technique, with a diamond tip, in the range from 4000 to 650 cm^−1^ at room temperature. The crystalline structure of ZnO and CS-ZnO films were tested by an X-ray diffractometer (Rigaku Dmax 2100, The Woodlands, TX, USA) with Cu Kα radiation (λ = 0.154 nm).

Impedance measurements were carried out using Agilent 4249 A in the frequency range 40 Hz–100 MHz with an amplitude of AC voltage 100 mV at room temperature. DC resistance *R* and capacitance *C* at the limit of zero frequency were calculated from fitting impedance spectra using ZView program. Conductivity and static dielectric constant (at the limit of zero frequency) were calculated from the following relationship: σ = d/(R∙S), *ε* = (C·d)/(*ε_0_*·S), where d and S are the thickness and area of samples, respectively. Film thickness was measured in each sample using micrometer Mitutoyo with resolution 1 mkm. The mechanical test was performed on an Instron universal tensiometer material testing system (model TX2plus). Each composite film was cut with dimensions according to ASTM Standard D638-Epsilon. Each strip was held with a distance between clamps of 25 mm. The test was performed with the lower grip was fixed, and the upper grip rose at an extension rate of 1 mm/sec at room temperature. All the failures occurred in the middle region of the testing strips. This test was repeated six times for each specimen to confirm its repeatability.

The measurements of ferroelectric polarization loops (P versus E) and deformation curves as a function of the applied field (butterfly curves) were obtained simultaneously by placing the samples in a measuring cell with parallel electrodes immersed in silicone oil to avoid dielectric breakage of the surrounding medium with voltage step 100 V before sample breakdown. The polarization and deformation curves presented in this work correspond to the maximum applied voltage measured before film breakdown. The ferroelectric measurements were based on the principle of the Sawyer-Tower circuit using a Precision LC materials analyzer, Radiant Technologies Inc., coupled with a TREK Model 609E-6 voltage amplifier source. Results presented in this work correspond to the maximum voltage before the breakdown of the sample.

## 3. Results

XRD analysis can supply information about the crystalline structure of ZnO NPs and Cs-ZnO NPs films (Figure 1). XRD pattern of neat CS and CS-ZnO (with 20 wt.% of NPs) show the diffraction peak at 2θ ≈ 24° (hydrate crystalline phase, Form 1) and a weak peak at 2θ ≈ 16.9° (hydrate crystalline phase, Form 2) [45]. The peaks observed in ZnO NPs and CS-ZnO films were in good agreement with the database of hexagonal ZnO particles (JCPDS No. 36-1451) and the results are reported in refs. [15,19,23]. This means that structure of ZnO NPs was not modified by the presence of CS [18,20], but the intensity of broad CS peak at 2θ ≈ 24° decrease in CS-ZnO films indicated the increase in the degree of amorphous regions of the nanocomposites films due to the interaction between the CS matrix and ZnO NPs and decreasing of water content [25]. Measurements of CS-ZnO films were carried out on copper substrate; therefore, there are additional diffraction patterns associated to Cu substrate at 2θ ≈ 43.3° and 2θ ≈ 50.4°; this fact is indicated in Figure 1.

The average size of particles (*D*) was calculated using the Debye–Scherrer equation [18,19]:(1)D=kλ/βcosθ
where the value of *k* is equal to 0.89, *λ* is the wavelength of X-ray (1.54°A), *β* is the full width at half maximum, and *θ* is the half of the diffraction angle. The average value of crystallites (calculated using three diffraction peaks (100), (002), and (101)) were 45.3 nm which correlates well with the dimension of NPs (*ca*. 40 nm).

Figure 2 shows the FTIR spectra of neat CS films and CS-ZnO films with 10 and 20 wt.% of ZnO NPs. In the case of chitosan, the broadband characteristic peak centered at 3252 cm^−1^ corresponds to the overlap of stretching vibration of –NH and –OH groups shift in CS-ZnO films to lower wavenumber at 3227 cm^−1^. The absorption peaks of CS at 1636 (amide I group), 1542 cm^−1^ (bending vibrations of NH_3_), and 1065 (the stretching vibration of C–O–C of the glycosidic linkage) in CS-ZnO films shift to lower wavenumber (Figure 2) due to the interaction of these group with ZnO and formation of a hydrogen bond between ZnO and chitosan. This result is in good agreement with previous reports [15,18,19,23,46].

Another confirmation of strong interaction between side groups of CS with ZnO NPs can be obtained from TGA measurements (Figure 3). It was previously reported [47] that neat CS exhibits a two-step weight loss. From room temperature to ca. 150 °C, the weight loss is related to the water evaporation and in the temperature range of 170–300 °C, the weight loss is due to the degradation of the CS [47]. Water absorption in CS is closely linked to the availability of amino and hydroxyl groups of CS that interact via hydrogen bonding with water molecules [47,48]. It has been observed that the water content depends upon ZnO NPs concentration (Figure 3) and it decreases with increasing weight% of NPs (11.7% in neat CS and 7.3% in CS-ZnO film with 20 wt.% of NPs, at the temperature 140 °C). As it was shown by FTIR analysis, these groups can bond with ZnO NPs; therefore, a decrease in the water absorption ability with increasing ZnO concentration is observed.

The results obtained from XRD, FTIR, and TGA measurements have shown an interaction between CS matrix and ZnO NPs that play an important role in the explanation of electrical and mechanical properties of the nanocomposite.

Figure 4a shows the dependence of DC conductivity and Figure 4b shows the dependencies of the dielectric constant in the limit of zero frequency as a function of ZnO NPs wt.%. It is evident from Figure 4b that dependence of dielectric constant exhibits a maximum at a concentration of ZnO NPs of ca. 15 wt.% and it is higher than the static dielectric constant of neat ZnO (*ca*. 8.5 [27]). It is noteworthy that the conductivity of CS-ZnO nanocomposite (Figure 4a) is sufficiently lower than the ZnO conductivity (*ca*. 1 × 10^−5^ S/cm [28,29]) and it decreases with ZnO NPs wt.%.

Similarly, to the dielectric constant behavior, there is maximum in the dependence of Young’s modulus on ZnO wt.% (Figure 5). Young’s modulus increases from 1.7 GPa (in neat CS) to 9.09 GPa in films with 15 wt. = % of NPs. At concentration of ZnO NPs 20 wt.%, Young’s modulus decreases to 4.5 GPa. The increasing of Young’s modulus with ZnO concentration has been previously reported [12,15,20] and it has been interpreted by an additional energy-dissipating mechanism [12], weakness of intermolecular hydrogen bonds of CS formation of new hydrogen bonds between CS and ZnO [15]. CS-PVA-ZnO NPs membrane study [25] reported the maximum in Young’s modulus at 10 wt.% of NPs which has been explained by the interaction of ZnO NPs with CS-PVA functional groups.

ZnO is a well-known material that exhibits both ferroelectric and piezoelectric behavior. Therefore, it is important to investigate these properties in CS-ZnO nanocomposites which can find applications in flexible electronics.

Figure 6 shows the ferroelectric hysteresis curve obtained in the CS-ZnO NPs film with 15 and 20 wt.% of NPs. The shape of the curve is typical for samples with electrical leakage, which prevents reaching saturation in the polarization [49]. This leakage current can be associated with the intrinsic proton conductivity of the CS matrix [50]. The corresponding deformation curves do not present symmetrical shape, probably because of the interaction between ZnO NPs and CS. The piezoelectric coefficient d_33_ was evaluated in the linear region of the deformation curve vs. applied voltages using the expression d_33_ = Δl/ΔV [51,52]. The piezoelectric coefficient d_33_ in CS-ZnO nanocomposite films with 15 wt.% of NPs (d_33_ = 65.9 *pC/N*) is higher than in neat ZnO NPs (between 0.4 and 12.4 *pC/N* [53,54]) and in poly(vinylidene fluoride) PVDF-ZnO flexible films (13.42 *pC/N* [55], 18.3 *pC/N* [56], 50 *pC/N* [52]) and compared with PVDF-PTTE-ZnO nanorods (70.3 *pC/N* with 15 wt.% of nanorods [57]). Note, that PVDF is a polymer with piezoelectric properties.

Such a high piezoelectric coefficient can be related to elastic properties of the CS matrix because the viscous and elastic properties play an important role in the piezoelectric performance of piezoelectric polymer composites [58].

Additionally, piezoelectric coefficient d_33_ demonstrate higher value (d_33_ = 65.9 *pC/N*) in CS-ZnO films with 15. wt.% of NPs than in films with 20 wt.% (d_33_ = 60.6 *pC/N*).

## 4. Discussion

Bulk conductivity of ZnO is ca. 10^−4^–10^−5^ S/cm and it depends upon intrinsic defects created by oxygen vacancies [28,29,30]). Conductivity of nanoparticles depends upon grain size, morphology, and microstructure and it is ca. 1.5 × 10^−7^ S/cm [59]. Because of the low volume fraction of ZnO NPs in CS-ZnO films (from 0 to 0.15 wt.%; see below the volume fraction calculation), the effective conductivity of nanocomposites practically depends upon the conductivity of neat CS (*ca*. 10^−7^ S/cm). The conductivity of neat CS is related to the Grotthuss mechanism in which the protons are originated from the protonated amino groups that can move along the hydrated molecule in the hydrogen-bonding network via hopping process [50]. Because of the strong interaction of reactive CS side groups with ZnO NPs (as probed by FTIR measurements) the number of generated protons and the number of hydrated molecules decrease (TGA measurements); this plausible scenario is responsible for the decreasing of nanocomposite’s conductivity.

As a rule, the effective dielectric constant *ε* and conductivity σ of a mixture of two materials with different *ε* and σ can be calculated using models as the Maxwell, Bruggeman, Lichtenecker, or different percolation model [60,61,62]. However, all these models produce a monotonic decreasing of the effective dielectric constant with increasing concentration of nanoparticles in the polymer (because *ε* of CS films is ca. 25 and *ε* of ZnO is ca. 8.5 [27]); it also can show a monotonic increasing of effective conductivity with increasing concentration of (because σ of CS is ca. 2 × 10^−7^ S/cm and σ of ZnO is ca. 10^−4^–10^−5^ S/cm).

In contrast to those models, refs. [63,64] proposed a three-phase model to describe the dielectric properties of polymer–ceramic composites. Here, the effective dielectric constant of such composite materials depends upon the *ε* of the polymer matrix, the *ε* of fillers, and the *ε* of interfacial layer between filler and the dielectric matrix. To describe such three-phase system, refs. [63,64] introduce a parameter *K*, termed the interfacial volume constant, which accounted for the matrix–filler interaction strength as:(2)$$\Phi_{int}=K\Phi_{NPs}\Phi_{pol}$$
where *Φ_int_, Φ_NPs_*, and *Φ_pol_* are the volume fractions of interfacial phase, dielectric particles, and polymer, respectively. *K* depends upon the degree of particle clustering.

In the case if dielectric constant of interfacial layer is higher than dielectric constant of polymer matrix and fillers,
*K* > 0, *ε_interfacial_* > *ε_polymer_, ε_interfacial_* > *ε_filler,_*(3)
with increasing of NPs concentration effective dielectric constant of nanocomposite increases, as observed in Figure 4b.

This model has demonstrated that the dependence of the dielectric constant on NPs concentration is nonmonotonic and can exhibit a maximum as a function of NPs concentration. This maximum appears when there is an overlap of interfacial layers due to NPs agglomeration, thus reducing the interfacial volume fraction that effectively decreases the effective value of dielectric constant of nanocomposite.

In this work, we experimentally fit obtained values of dielectric constant *ε* in CS-ZnO NPs films by equations proposed in refs. [63,64] using the Scilab program. The least-squares fitting was performed using standard genetic algorithm optimization functions in the Scilab [65] numerical computational package. The dielectric constant of CS was obtained from measurements on neat CS and *ε* of ZnO was taken 8.5 [27]. Only the values of K and *ε* interfacial parameters are the adjustable parameters.

To convert weight fraction (Wt) to volume fraction (V) of the ZnO NPs, the next equation can be used [66]:(4)$$V=\frac{W_{t}}{W_{t}+(\rho_{ZnO}/\rho_{CS}\left(1-W_{t}\right)}\;$$
where, *ρ*_*ZnO*_ and *ρ*_*CS*_ denote the *ZnO* and *CS* density.

The density of CS films is ca. 1.5 g cm^−3^ [67,68], and the true density of ZnO is 5.6 g cm^−3^ [3].

As a result of optimization, the fitted values are *K* = 18.3, and interface dielectric constant equals 69.9. The results of the referred fittings are shown on inset of Figure 4b as a continuous line. One can see that this three-phase model fits well the experimental results by predicting a maximum in the dielectric constant. Positive value of K means that there are significant interfacial interactions between CS and ZnO NPs; these observations were confirmed by FTIR and TGA measurements. Additionally, an interfacial layer dielectric constant value of 69.9 is higher than that of CS and ZnO NPs which is responsible for the observed maximum in Figure 4b. In summary, the three-phase model is able to capture the correct physics of the nanocomposite by corroborating the behaviors and trends of the experimental measurements.

Similarly, a maximum at 15 wt.% of ZnO is observed as in the dependency of Young’s modulus on ZnO concentration (Figure 5). According to refs. [69,70], the Young’s modulus in CS films increases with decreasing of water content due to plasticizing effect of water and change in the glass transition temperature. In the TGA measurements water content decreases ca. 5% in CS-ZnO films with 20 wt.% of NPs when compared with neat CS. Based upon the results reported in ref. [69,70], this decreasing of water content corresponds to the increasing of Young’s modulus approximately to 0.5–1 GPa. Therefore, all contributions in elastic module can be related to the change of NPs concentration. Furthermore, a maximum at 15 wt.% of ZnO is observed as in the dependency of Young’s modulus and dielectric constant on ZnO concentration (Figure 4b and Figure 5). The explanations proposed in the literature on the change of Young’s modulus in ZnO nanocomposites are the following: An additional energy-dissipating mechanism [12], the weakness of intermolecular hydrogen bonding of CS, or the formation of new hydrogen bonding between CS and ZnO [15]; these explanations cannot be directly applied to properly address the existence of a maximum in both dielectric and mechanical properties. According to refs. [71,72,73], the mechanical properties exhibit a similar dependency on the formation of an interfacial layer. Above the percolation threshold, the interfacial regions surrounding the ZnO NPs overlap indicating percolation in the clusters which dominates in the mechanical properties [72]. Because of the agglomeration of NPs, there is a maximum in both dielectric constant and Young’s modulus because further agglomeration of NPs tends to destroy the interfacial regions.

Now let us discuss the dependence of the piezoelectric coefficient on the concentration of NPs. There have been several reports that proposed different models, for instance Refs. [58,74,75,76] reported an increase of d_33_ with the volume fraction of the piezoelectric NPs. Moreover, in polymer nanocomposites with NPs with high dielectric constant (such as PZT, BaTiO_3,_ BZT) there is an increase of d_33_ with an increase of the dielectric constant with NPs content [74,75,76]. However, in BaTiO_3_-epoxy-ZnO (with a fixed concentration of BaTiO_3_) [77] and PVDF-ZnO [56], the dependences of the piezoelectric coefficient and the dielectric constant on the concentration of ZnO exhibit a maximum (similarly to the maximum presented in Figure 4b). This maximum cannot be explained by classical conductivity percolation effect because the conductivity of CS-ZnO films decreases with ZnO NPs wt.% (Figure 4a). However, the proposed three-phase model describes well the maximum in dielectric constant (an overlap of interfacial layers due to NPs agglomeration) that effectively decreases the value of dielectric constant and most likely the piezoelectric coefficient. A similar hypothesis on the decreasing of d_33_ due to NPs agglomeration (based on the SEM measurements in BaTiO_3_-epoxy-ZnO) is proposed in ref. [75].

The piezoelectric coefficient d_33_ in CS-ZnO nanocomposite films with 15 wt.% of NPs (d_33_ = 65.9 *pC/N*) is higher than most of polymer-ZnO nanocomposites (see Table 1) and it compares with PVDF-PTTE-ZnO nanorods (70.3 *pC/N*).

It is well-known that in classic ferroelectric materials such as BaTiO_3_, a significant increase in the piezoelectric coefficient in materials with nanodomain structure was observed. The value of d_33_ in samples with grains 50 nm increases more than twice to 416 pC/N [81] compared with grains of 500 nm (200 pC/N). In the case of the ZnO NPs with dimension ca. 40 nm the size of the nanodomains must be less than 40 nm that can increase d_33_ in CS-ZnO nanocomposite. Furthermore, high value of d_33_ in CS-ZnO nanocomposite films can be related to their elastic properties that play an important role in the piezoelectric performance of piezoelectric polymer composites [58]. Additionally, refs. [82,83] reported that CS films with 91.2% of deacetylation degree exhibit a piezoelectric coefficient d_33_ between 7 and 18.4 *pC/N*. These piezoelectric properties are observed because of the fact that crystalline part of CS has non-centrosymmetry orthorhombic structure with piezoelectric properties [82]. However, in our work we did not observe piezoelectric properties of neat CS films; this observation may be traceable to the lower deacetylation degree of CS (*ca.* 72%). Nevertheless, in the presence of an electrical field (at which we measured d_33_) an alignment of CS chains can be observed. The alignment of CS chains can be correlated with the increase in dielectric and piezoelectric properties of the composites [84]. Therefore, the high d_33_ in CS/ZnO films can be traced to the synergistic effect of nanodomain structure of NPs, piezoelectricity of CS, elastic properties of films, and optimum NPs concentration.

## 5. Conclusions

The investigation of CS-ZnO films with different wt.% of NPs shows that a maximum value is observed in both dielectric constant and Young’s modulus. Similarly, at the same concentration of NPs there appears the highest value of the piezoelectric coefficient. This maximum can be related to cluster agglomeration of ZnO NPs above the dielectric and mechanical percolation threshold (15 wt.% of ZnO NPs). These properties of nanocomposite’s films are interpreted by using a three-phase model which includes: (1) CS matrix, (2) ZnO NPs, and (3) interfacial layer between ZnO and CS matrix. This interface layer is responsible for the higher dielectric constant when compared with the *ε* of neat CS and ZnO, a higher Young’s modulus, and a higher d_33_. The piezoelectric coefficient d_33_ in CS-ZnO nanocomposite films with 15 wt. % of NPs (d_33_ = 65.9 *pC/N*) is higher than in the most of polymer-ZnO nanocomposites because of the synergistic effect of nanodomain structure of NPs, piezoelectricity of CS, elastic properties of CS, and optimum NPs concentration.

Based upon the presented methodology, one can try to fine-tune the desired properties by manipulating the concentration and agglomeration of NPs that ultimately control the molecular interactions. It is noteworthy that these variables not only depend upon the dimension of NPs but also on the method of preparation and the chemistry of constituents. However, the methodology presented here allows to determine the direct relationship between dielectric, mechanical, piezoelectrical properties, and the concentration of nanoparticles that may prove useful in the design and optimization of polymer-based nanocomposites for different applications.

In summary, the molecular understanding of nanoscale properties of CS-ZnO nanocomposites is relevant in the development of biocompatible sensors, actuators, nanogenerators, etc. for flexible electronics, and biomedical applications.

## Figures and Tables

**Figure 1 polymers-12-01991-f001:**
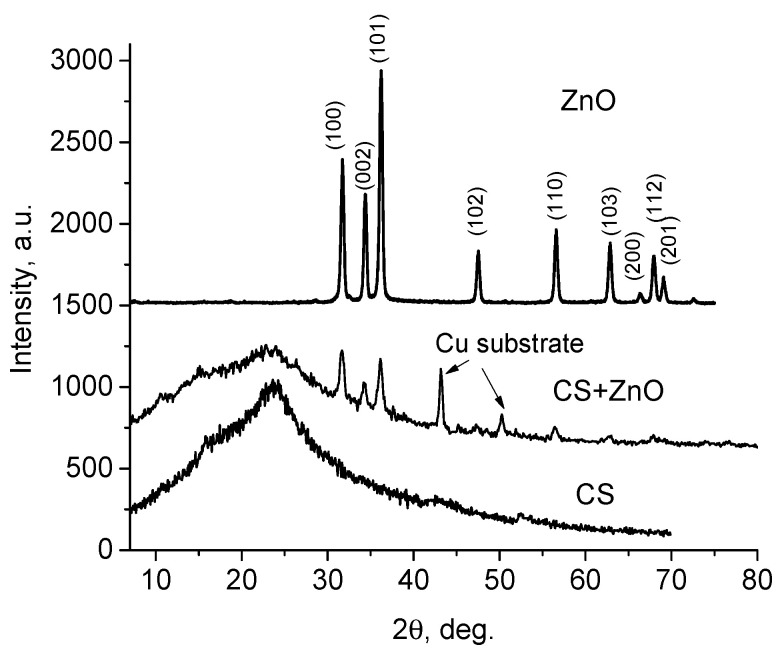
X-ray diffraction patterns of neat chitosan (CS) film, CS ZnO NPs composite, and ZnO nanoparticles (NPs).

**Figure 2 polymers-12-01991-f002:**
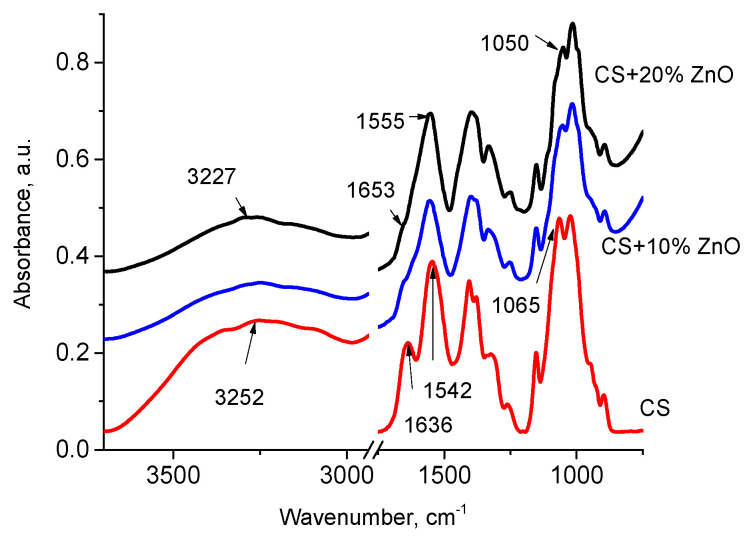
FTIR spectra of neat CS films and CS-ZnO films with 10 and 20 wt.% of ZnO.

**Figure 3 polymers-12-01991-f003:**
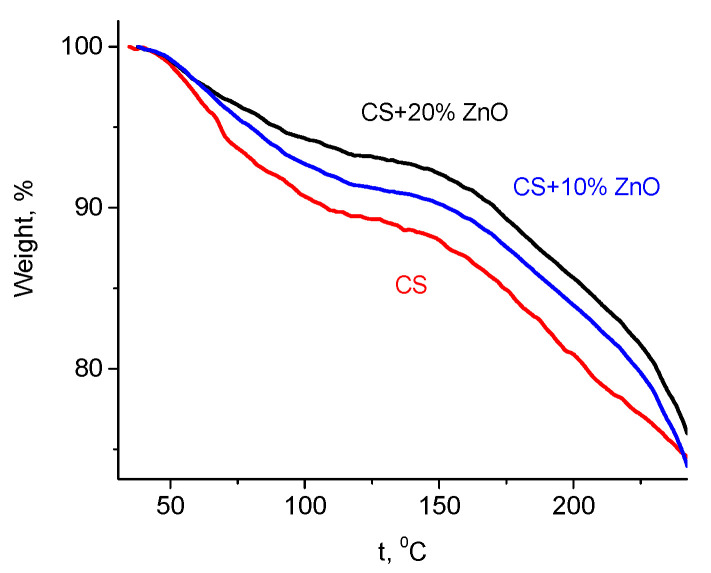
TGA measurements of pure CS and CS-ZnO NPs films with 10 and 20 wt.% of NPs.

**Figure 4 polymers-12-01991-f004:**
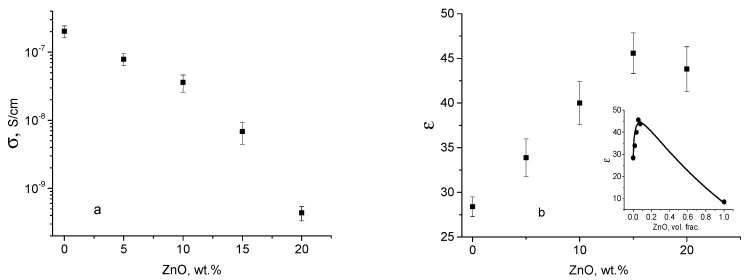
Dependences of (**a**) DC conductivity (σ) and (**b**) dielectric constant (*ε*) in the limit of zero frequency obtained in CS-ZnO films with different ZnO concentration at room temperature. Insert in Figure 4b shows dependence of *ε* on the volume fraction of ZnO NPs: points-experimental measurements and continuous line-results of the fitting.

**Figure 5 polymers-12-01991-f005:**
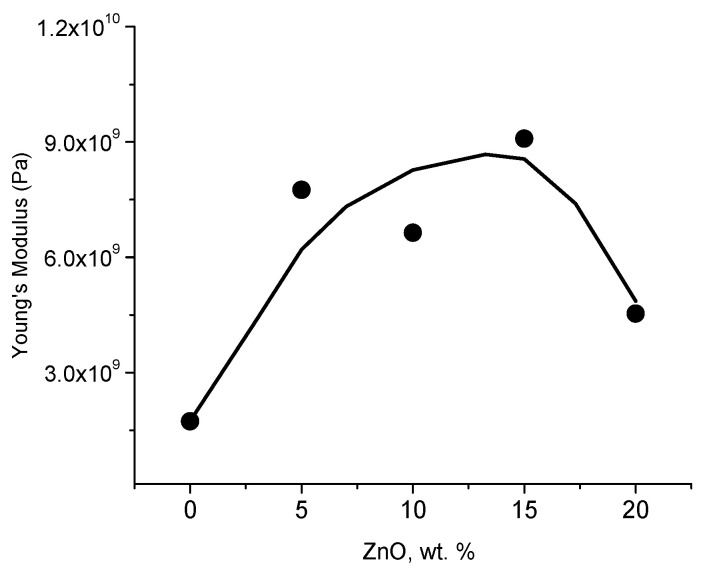
Dependence of Young’s modulus of CS ZnO NPs films with different wt.% of NPs (points). The continuous line is a guide to the eye.

**Figure 6 polymers-12-01991-f006:**
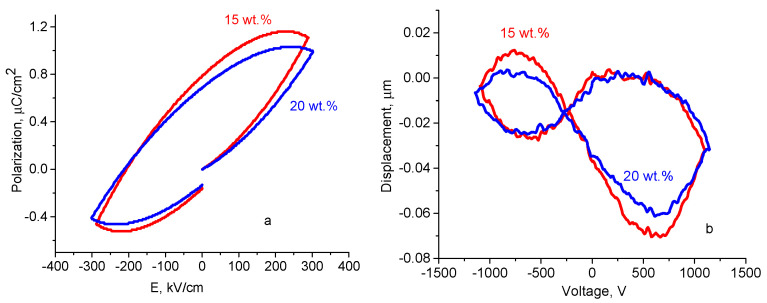
Ferroelectric hysteresis curve of (**a**) CS-ZnO NPs nanocomposites, (**b**) corresponding deformation curve Δl vs. V for CS-ZnO films with different wt.% of NPs indicate on the graph.

**Table 1 polymers-12-01991-t001:** Comparison of piezoelectric d_33_ values in polymer nanocomposites with ZnO NPs.

Polymer Nanocomposite	NPs Dimension (nm)	d_33_ (*pC/N*)	Refs
ZnO, bulk	-	0.4–12.4	[53,54]
photo-epoxy/ZnO films	Less 100	15–23	[78]
PVDF/ZnO films	50–80	13.42	[55]
PVDF/ZnO nanoporous films	35–45	18.3	[56]
PVDF β-phase/ZnO	50–150	50	[54]
PVDF-PTTE/ZnO nanorods	-	70.3	[57]
PHB/ZnO scaffolds	80–100	13.7	[79]
PVDF/ZnO nanorods	-	−1.17	[80]
CS/ZnO films	40	65.9	This work

Note that PVDF is a polymer with piezoelectric properties.

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
