# Peer review of "Chitosan-ZnO Nanocomposites Assessed by Dielectric, Mechanical, and Piezoelectric Properties"

_polymers, 2020, doi:10.3390/polym12091991_

Round 1

Reviewer 1 Report

see attachment

Author Response

Response to Reviewer 1 Comments

COMMENTS

The understanding of nanoscale CS/ZnO nanocomposites is important in the

development of actuators, nanogeneretors for flexible electronics. Many work have investigated the similiar nanocomposites. Compared with previous literature, in addition to dielectric constant and Young’s modulus, this paper just reports the influence of ZnO content on the conductivity of CS-ZnO nanocomposite and exhibits many technical errors.

  1. In Figure 1, the XRD curves of CS-ZnO show some new peaks at 43o and 51o. Why?

Response 1

We measured CS-ZnO films on Cu substrate, therefore these lines at 2θ≈43.30 and 2θ≈50.40 correspond to diffraction from Cu substrate.

We also modified the manuscript as follows:

“Measurements of CS-ZnO films were carried out on copper substrate; therefore, there are additional diffraction patterns associated to Cu substrate at 2θ≈43.30 and 2θ≈50.40. this fact is indicated on Fig. 1.”

  1. In Figure 2, one cannot see the peaks at 3252 cm-1 and 3227cm-1 as the author claimed.

Response 2

On Fig. 2 we have shown breaks on X-axis that readers can better see the shift of CS peaks due to interaction with ZnO NPs. Note that if X-axis (see above) is not broken, the associated peaks cannot be observed in detail; full X-axis resolution would look similar to previous works [15, 18, 19, 23]. 

We changed the manuscript as follows

“In the case of chitosan, the broadband characteristic peak centered at 3252 cm-1 corresponds to the overlap of stretching vibration of –NH and –OH groups; when strong molecular interaction occurs in the presence of ZnO the stretching vibration shifts to lower wavenumber at 3227 cm-1”.

  1. In Figure 3, the lower loss of water in TGA curves is attributed to the increased OHbonds with ZnO particles. How does it agree with the lower elastic modulus for 20%ZnO composition as presented in Figure 5?

Response 3

According to refs [69,70] Young’s modulus in CS increases with decreasing of water content due to plasticizing effect of water and its direct effect on the glass transition temperature. According to TGA measurements water content decreases ca. 5% in CS-ZnO films with 20 wt.% of NPs when compared with neat CS. Based upon the results reported in ref. [69,70] this decreasing of water content corresponds to the increasing of Young’s modulus to 0.5-1 GPa. Consequently, all contributions to elastic module can be related to the change of NPs concentration.

We added in the text:

“According to refs [69,70] Young’s modulus in CS increases with decreasing of water content due to plasticizing effect of water and its direct effect on the glass transition temperature. In the  TGA measurements water content decreases ca. 5% in CS-ZnO films with 20 wt.% of NPs when compared with neat CS. Based upon the results reported in ref. [69,70] this decreasing of water content corresponds to the increasing of Young’s modulus to 0.5-1 GPa. Consequently, all contributions to elastic module can be related to the change of NPs concentration.  Furthermore, a maximum in both Young’s modulus (Fig. 5) and dielectric constant (Fig. 4b) is observed at 15 wt.% of ZnO. According to refs. [66–68] the mechanical properties exhibit a similar dependence on the interfacial layer; after the percolation threshold, the interfacial regions surrounding the ZnO NPs overlap indicating percolation in the clusters which dominate the mechanical properties [67]”.  

  1. Line 183. “The increasing of Young’s modulus with ZnO concentration has been

observed in [12,15,20] and has been interpreted by an additional energy-dissipating mechanism [12], weakness of intermolecular hydrogen bonds of CS formation of new hydrogen bonds between CS and ZnO [15].” The author just cited other papers to explain, lacking of self opinion.

Response 4.

At line 183 we wrote the explanation reported in the literature. However, in the Discussion Section we provided our interpretation of results.

“Similarly, a maximum at 15 wt.% of ZnO is observed as in the dependency of Young’s modulus on ZnO concentration (Fig. 5). According to refs. [66–68] the mechanical properties exhibit a similar dependency on the interfacial layer. After the percolation threshold, the interfacial regions surrounding the ZnO NPs overlap indicating percolation in the clusters which dominates in the mechanical properties [67]. Therefore, due to the agglomeration of NPs at the same concentration observed maximum in dielectric constant and Young’s modulus”.

To clarify this question, we added in the manuscript the following changes:

“According to refs [69,70], the Young’s modulus in CS films increases with decreasing of water content due to plasticizing effect of water and change in the glass transition temperature. In the TGA measurements water content decreases ca. 5% in CS-ZnO films with 20 wt.% of NPs when compared with neat CS. Based upon the results reported in ref. [69,70], this decreasing of water content corresponds to the increasing of Young´s modulus approximately to 0.5-1 GPa. Therefore, all contributions in elastic module can be related to the change of NPs concentration. Furthermore, a maximum at 15 wt.% of ZnO is observed as in the dependency of Young’s modulus and dielectric constant on ZnO concentration (Figs. 4b, 5). The proposed in the literature explanations on the change of Young’s modulus in ZnO nanocomposites are the following: an additional energy-dissipating mechanism [12], the weakness of intermolecular hydrogen bonding of CS or the formation of new hydrogen bonding between CS and ZnO [15]; these explanations cannot be directly applied to properly address the existence of a maximum in both dielectric and mechanical properties. According to refs. [71-73], the mechanical properties exhibit a similar dependency on the formation of an interfacial layer. Above the percolation threshold, the interfacial regions surrounding the ZnO NPs overlap indicating percolation in the clusters which dominates in the mechanical properties [72]. Due to the agglomeration of NPs, there is a maximum in both dielectric constant and Young’s modulus because further agglomeration of NPs tends to destroy the interfacial regions.

  1. Figure 4b. is not discussed in the text.

Response 5.

All paper based on explanation of anomalous behavior of dielectric constant on NPs concentration with maximum (Fig. 4b) and fitting of this dependence using three phase model (insert on Fig. 4b). Therefore, we cannot agree that Fig. 4b is not discussed in the text.

We modified the manuscript as follows:

“Fig. 4a shows the dependence of DC conductivity and Fig. 4b shows the dependencies of the dielectric constant in the limit of zero frequency as a function of ZnO NPs wt. %. It is evident from Fig. 4b that dependence of dielectric constant exhibits a maximum at a concentration of ZnO NPs of ca. 15 wt. % and it is higher than the static dielectric constant of neat ZnO (ca. 8.5 [27]). It is noteworthy that the conductivity of CS-ZnO nanocomposite (Fig. 4a) is sufficiently lower than the ZnO conductivity (ca. 1 10-5 S/cm [28,29]) and it decreases with ZnO NPs wt.%”.

  1. Line 220 “refs. [60,61] proposed a three-phase model to describe the dielectricproperties of polymer-ceramic composites.” There is no original innovation in thismodel.

Response 6.

We thank the Reviewer for this comment; we believe we did not get across this issue properly.

It is noteworthy that according to ScopusR the use of the referred three-phase model has been cited in 152 publications; however, these papers only suggest the possibility of applying the three-phase model polymer-ZnO NPs composites in 2 review articles. No attempt has been offered to fit any experimental data yet.

In this regard, our group has offered for the first time, the application of this model to fully explain the anomalous dependence of dielectric constant in CS-ZnO nanocomposites. To accomplish this task, we have also developed a computer program to fit our experimental data. It is noteworthy that as a rule, the effective dielectric constant ε and conductivity σ of a two-component mixture with different ε and σ can be calculated by using models such as the Maxwell, Bruggeman, Lichtenecker and other classical percolation models. In the literature, there are on the order of thousand papers which used these models. However, the use of such classical models fails to capture the correct physics due to low conductivity of CS-ZnO and  the application of the three-phase model allows correct describe properties of CS-ZnO nanocomposites.

  1. Figure 6 shows the lossy behavior of the composite samples. The asymmetric

butterfly strain behavior is indicative of impurity effect in the samples. The

piezoelectric property has no sound basis for use.

Response 7.

We agree with the Referee that asymmetric butterfly behavior can be related to impurities; however, our XRD measurements do not reveal any impurity in our nanocomposites (see Fig. 1). In this regard, the presence of asymmetric butterfly behavior can be traced to the presence of non-1800 domain reorientation (J. Korean. Phys. Soc., 57, 2010, 902); the existence of remnant displacement after the removal of electrical field (2018 IEEE Conference on Decision and Control, 2018); the incomplete domain switching (Appl. Phys. Lett. 116, 242902, 2020), among others.

We respectfully disagree with the Referee that piezoelectric phenomenon has no sound basis in our study. In this regard, the literature offers vast studies on the referred asymmetric behavior for applications in shape (Ferroelectrics, 368:185–193, 2008) and strain memory (J. Korean. Phys. Soc., 57, 2010, 902) for skin regeneration (Colloids and Surfaces B: Biointerfaces, 2020) etc. Other published works deal with the theoretical and experimental study of non-standard, asymmetric butterfly loops for the development of new application of piezoelectric materials.

This paper is lack of innovation and academic accuracy. Its analysis is too simple and

superficial, so it is recommended to be refused.

Response 8.

We regret the perception of the Referee on the quality of our manuscript; however, we feel we have provided sufficient evidence and discussion in the new version of the manuscript as well in this response to properly address the innovation on the study of chitosan-ZnO nanocomposites. We believe our study not only innovates on capturing the correct physics of low conductivity nanocomposites but also on the possibility of triggering new relevant technological applications

We now address in more detail other points raised by the Referee. The Referee wrote in his/her review the following statement: “Compared with previous literature, in addition to dielectric constant and Young’s modulus, this paper just reports the influence of ZnO content on the conductivity of CS-ZnO nanocomposite”. This assertion is not entirely accurate because we performed a thorough literature review of 84 references that are not only related to CS-ZnO composites but also to most reported polymer nanocomposites. We reviewed the correlation between conductivity and dielectric constant; dielectric constant and elastic modulus; dielectric constant and piezoelectric coefficient. It is noteworthy that those reported correlations proposed a variety of explanations based upon limited number of measurements that often lead to the incomplete physics of the phenomena. In this regard, we believe our study sheds light on the nanoscale properties of CS/ZnO nanocomposites by proposing a correlation between dielectric constant, Young’s module, piezoelectric coefficient and conductivity which has been confirmed by experimental measurements and by simulation. The methodology used in our study can be readily extended to the investigation and optimization of other polymer-dielectric fillers nanocomposites for different applications. In summary, we believe that our work is new and will be interesting to the scientific community.

Reviewer 2 Report

This paper studied the dielectric and piezoelectric properties of chitosan-ZnO nanocomposites. The largely increased dielectric constant and piezoelectric coefficient are interesting. However, the experimental results are not discussed in-depth. So the manuscript should be revised to add more in-depth discussion. Following are the detailed suggestions.

1, Figure 4, neat ZnO has higher conductivity and lower dielectric constant compared with CS. Why the CS-ZnO nanocomposites show decreased conductivity and increased dielectric constant with the increase of ZnO content?

2, Also, the increase of the dielectric constant is apparent, how to explain this phenomena?

3, The Young’s modulus of the CS-ZnO nanocomposites increases about 5 times compared with neat CS, which is much higher than the results reported in literature. This should be discussed.

4, How is the d33 calculated?

5, The greatly increased d33 in CS-ZnO should be comprehensively discussed.

Author Response

Response to Reviewer 2 Comments

Comments and Suggestions for Authors

This paper studied the dielectric and piezoelectric properties of chitosan-ZnO nanocomposites. The largely increased dielectric constant and piezoelectric coefficient are interesting. However, the experimental results are not discussed in-depth. So the manuscript should be revised to add more in-depth discussion. Following are the detailed suggestions.

We agree with comments of referee 2; we have modified the discussion section and added additional information.

1, Figure 4, neat ZnO has higher conductivity and lower dielectric constant compared with CS. Why the CS-ZnO nanocomposites show decreased conductivity and increased dielectric constant with the increase of ZnO content?

Response 1

 We added the following explanation of conductivity

“Bulk conductivity of ZnO is ca. 10-4- 10-5 S/cm and it depends upon intrinsic defects created by oxygen vacancies [28,29,30]). Conductivity of nanoparticles depends upon grain size, morphology and microstructure and it is ca. 1.5 10-7 S/cm [59]. Due to the low volume fraction of ZnO NPs in CS-ZnO films (from 0 to 0.15 wt.%; see below the volume fraction calculation). Therefore, the effective conductivity of nanocomposites practically depends upon the conductivity of neat CS (ca. 10-7 S/cm). The conductivity of neat CS is related to the Grotthuss mechanism in which the protons are originated from the protonated amino groups and can move along the hydrated molecule in the hydrogen-bonding network via hopping process [50]. Due to the strong interaction of reactive CS side groups with ZnO NPs (as probed by FTIR measurements) the number of generated protons and the number of hydrated molecules decrease (TGA measurements); this plausible scenario is responsible for the decreasing of nanocomposite’s conductivity”.

We also added the following about the increase of the dielectric constant with the increase of ZnO content:

“In contrast to those models, refs. [60,61] proposed a three-phase model to describe the dielectric properties of polymer-ceramic composites. Here, the effective dielectric constant of such composite materials depends upon the ε of the polymer matrix, the ε of fillers, and the ε of interfacial layer between filler and the dielectric matrix. To describe such three-phase system, refs. [60,61] introduced a parameter K, termed the interfacial volume constant, which accounted for the matrix-filler interaction strength as:

 (),

where Φint, ΦNPs, and Φpol are the volume fractions of interfacial phase, dielectric particles, and polymer, respectively. K depends upon the degree of particle clustering.

In the case if dielectric constant of interfacial layer is higher than dielectric constant of polymer matrix and fillers,

K>0, εinterfacialpolymer, εinterfacial > εfiller,   ()

with increasing of NPs concentration effective dielectric constant of nanocomposite will increases, that observed on Fig. 4b.

This model has demonstrated that the dependence of the dielectric constant on NPs concentration is nonmonotonic and can exhibit a maximum as a function of NPs concentration. This maximum appears when there is an overlap of interfacial layers due to NPs agglomeration, thus reducing the interfacial volume fraction that effectively decrease the effective value of dielectric constant of nanocomposite”.

2, Also, the increase of the dielectric constant is apparent, how to explain this phenomena?

Response 2

We have explained in detail the increasing of the dielectric constant above, in brief:

For the case where the dielectric constant of interfacial layer is higher than the dielectric constant of polymer matrix and fillers, the increase of NPs concentration promotes an increase of the effective dielectric constant of nanocomposite as observed in Fig. 4b.

3, The Young’s modulus of the CS-ZnO nanocomposites increases about 5 times compared with neat CS, which is much higher than the results reported in literature. This should be discussed.

Response 3

Our results do not significatively differ from data reported in the literature. For example, in ref. [12] Young’s modulus increases from 113.7 MPa for neat CS to 924.6 MPa in CS-ZnO with 10 wt. % of NPs; in ref. [20] the elastic modulus E changes from 15 GPa to 45 GPa for composites with 15 wt.% of NPs. In contrast, Ref. [J.  Polymers and the Environment (2020) 28:1216–1236) reports an E increase from 2 GPa to 2.2 GPa with 6.5 wt. % of NPs; and in CS-PVA-ZnO [Colloid&Surf. A, 2019, 123821] reports an E increase from 4.6 GPa in neat CS-PVA to 5.75 GPa with 10 wt.% of ZnO. We propose that such dispersion on the values of the Young’s modulus can be related to good or bad NP dispersion. In this regard, our work reveals we have achieved good dispersion as in Refs. [12, 20].

We added in the text next discussion about Young’s modulus of the CS-ZnO nanocomposite:

“According to refs [69,70], the Young’s modulus in CS films increases with decreasing of water content due to plasticizing effect of water and change in the glass transition temperature. In the TGA measurements water content decreases ca. 5% in CS-ZnO films with 20 wt.% of NPs when compared with neat CS. Based upon the results reported in ref. [69,70], this decreasing of water content corresponds to the increasing of Young´s modulus approximately to 0.5-1 GPa. Therefore, all contributions in elastic module can be related to the change of NPs concentration. Furthermore, a maximum at 15 wt.% of ZnO is observed as in the dependency of Young’s modulus and dielectric constant on ZnO concentration (Figs. 4b, 5). The proposed in the literature explanations on the change of Young’s modulus in ZnO nanocomposites are the following: an additional energy-dissipating mechanism [12], the weakness of intermolecular hydrogen bonding of CS or the formation of new hydrogen bonding between CS and ZnO [15]; these explanations cannot be directly applied to properly address the existence of a maximum in both dielectric and mechanical properties. According to refs. [71-73], the mechanical properties exhibit a similar dependency on the formation of an interfacial layer. Above the percolation threshold, the interfacial regions surrounding the ZnO NPs overlap indicating percolation in the clusters which dominates in the mechanical properties [72]. Due to the agglomeration of NPs, there is a maximum in both dielectric constant and Young’s modulus because further agglomeration of NPs tends to destroy the interfacial regions”.

How is the d33 calculated?

Response 4

We included in the text:

“The piezoelectric coefficient d33 of the BTO pellet was evaluated in the linear region of the deformation curve vs applied voltages using the expression d33 = Δl / ΔV [51,52]”.

  1. The greatly increased d33 in CS-ZnO should be comprehensively discussed.

Response 5

We included in the text the next explanation:

“The piezoelectric coefficient d33 in CS-ZnO nanocomposite films with 15 wt. % of NPs (d33=65.9 pC/N) is higher than most of polymer-ZnO nanocomposites (see Table 1) and it compares with PVDF-PTTE-ZnO nanorods (70.3 pC/N).

It is well known that in classic ferroelectric materials such as BaTiO3, was observed a significant increase in the piezoelectric coefficient, in material with nanodomain structure. The value of d33 in samples with grains 50 nm increases more than twice to 416 pC/N [81] compare with grains 500 nm (200 pC/N). In the case of the ZnO Nps with dimension ca. 40 nm the size of the nanodomains must be less than 40 nm that can increase d33 in CS-ZnO nanocomposite. Furthermore, high value of d33 in CS-ZnO nanocomposite films can be related to their elastic properties which  play an important role in the piezoelectric performance of piezoelectric polymer composites [58]. Additionally, refs.  [82,83] reported that CS films with 91.2 % of deacetylation degree exhibit a piezoelectric coefficient d33 between 7-18.4 pC/N.  These piezoelectric properties are observed due to fact that crystalline part of CS has non-centrosymmetry orthorhombic structure with piezoelectric properties [82]. However, in our work we did not observe piezoelectric properties of neat CS films; this observation may be traceable to the lower deacetylation degree of CS (ca. 72%). Nevertheless, in the presence of an electrical field (at which we measured d33) can be observe an alignment of CS chains.  The alignment of CS chains can be correlated with the increase in dielectric and piezoelectric properties of the composites [84]. Therefore, the high d33 in CS/ZnO films can be traced to the synergistic effect of nanodomain structure of NPs, piezoelectricity of CS, elastic properties of films and optimum NPs concentration.

We have also added new references in the text to support our answer on the referee reports.

Round 2

Reviewer 1 Report

not to my level of approval

Author Response

We have English-proof the entire manuscript and we have checked grammar and syntax accordingly.